# Generalized Bell Scenarios: Disturbing Consequences on Local-Hidden-Variable Models

**DOI:** 10.3390/e25091276

**Published:** 2023-08-30

**Authors:** André Mazzari, Gabriel Ruffolo, Carlos Vieira, Tassius Temistocles, Rafael Rabelo, Marcelo Terra Cunha

**Affiliations:** 1Instituto de Física Gleb Wataghin, Universidade Estadual de Campinas (Unicamp), Campinas 130830-859, Brazil; a194155@dac.unicamp.br (A.M.); g230334@dac.unicamp.br (G.R.); tassius.maciel@ifal.edu.br (T.T.); rlrabelo@unicamp.br (R.R.); 2Instituto de Matemática, Estatística e Computação Científica, Universidade Estadual de Campinas (Unicamp), Campinas 130830-859, Brazil; c261234@dac.unicamp.br; 3Department of Computer Science, The University of Hong Kong, Pokfulam Road, Hong Kong; 4Departamento de Física, Instituto de Ciências Exatas, Universidade Federal de Minas Gerais, Belo Horizonte 30123-970, Brazil; 5Instituto Federal de Alagoas-Campus Penedo, Rod. Eng. Joaquim Gonçalves-Dom Constantino, Penedo 57200-000, Brazil

**Keywords:** Bell nonlocality, contextuality, generalized Bell scenarios

## Abstract

Bell nonlocality and Kochen–Specker contextuality are among the main topics in the foundations of quantum theory. Both of them are related to stronger-than-classical correlations, with the former usually referring to spatially separated systems, while the latter considers a single system. In recent works, a unified framework for these phenomena was presented. This article reviews, expands, and obtains new results regarding this framework. Contextual and disturbing features inside the local models are explored, which allows for the definition of different local sets with a non-trivial relation among them. The relations between the set of quantum correlations and these local sets are also considered, and post-quantum local behaviours are found. Moreover, examples of correlations that are both local and non-contextual but such that these two classical features cannot be expressed by the same hidden variable model are shown. Extensions of the Fine–Abramsky–Brandenburger theorem are also discussed.

## 1. Introduction

Quantum theory is arguably the most successful scientific theory ever developed due to its remarkable predictions in a wide range of phenomena. Nevertheless, many of its features are counter-intuitive and still demand a better understanding. Among these features, two nonclassical phenomena that are particularly interesting are *Bell nonlocality* [1] and *Kochen–Specker (KS) contextuality* [2].

Both Bell nonlocality and KS contextuality are concerned with the possibility of explaining the results of quantum theory by means of *hidden variable models* in an effort to retrieve more intuitive descriptions of the observed results. The main difference between such concepts is the fact that the former is related to experiments involving two or more spatially separated systems, while the latter bypasses such a demand at the expense of demanding a priori compatibility relations between sets of measurements. In seminal works [3,4], it was shown that, for both phenomena, the predictions of quantum mechanics cannot be explained with such classical-like mechanisms, a fact that has been experimentally verified in several works [5,6,7,8,9,10,11,12,13].

Despite their similar motivations, the studies on Bell nonlocality and KS contextuality were developed through independent paths. In recent years, however, there have been different research programs that have proposed unified theoretical frameworks for both phenomena [14,15,16,17]. Among such proposals, the one in Ref. [18] deals with extensions of Bell scenarios that include the possibility of compatible measurements within each laboratory, allowing the investigation of Bell nonlocality and KS contextuality in well-defined experimental scenarios. In the same work, it was shown that this approach leads to new types of Bell inequalities, which were useful to witness the nonlocality of quantum states that could not be detected with standard Bell tests. Moreover, in Ref. [19], the same framework was applied to show that Bell nonlocality and KS contextuality could be concomitantly observed in the same experiment, thus proving that there is no monogamic relation between such concepts.

In this manuscript, we expand the ideas introduced in Ref. [18] and obtain several new results related to Bell nonlocality and KS contextuality. In particular, it is shown that this framework allows for the definition of different local sets, depending on the types of local strategies available for each party. The cornerstone of this generalization of Bell scenarios is that each separated party is allowed to measure local contexts; this opens possibilities to explore features related to contextuality and disturbance within the local model. One of our main results shows that there are correlations that are non-disturbing on the observational level but are local if and only if disturbance is allowed inside the local decomposition.

Moreover, different possibilities for sets of behaviours that are both local and non-contextual are also explored. Another interesting result is that there are correlations with a local decomposition for the joint behaviour and independent non-contextual decompositions for the marginal behaviours of each party, such that it is not possible to find a unified model that includes these two classical features. Extensions of the Fine–Abramsky–Brandenburger [20,21] Theorem are also considered, and it is shown that, in the framework used in this work, this is not the most general way to characterize behaviours that are both local and non-contextual. Finally, the set of quantum behaviours is defined, and some local behaviours that do not belong to it are presented.

We organized this work as follows. Section 2 reviews the concepts for standard Bell and KS contextuality scenarios; in Section 3, the concepts involving the framework of Bell scenarios with compatible measurements are reviewed and developed; Section 4 discusses the representation of the correlation sets as convex polytopes; the relation of quantum behaviours with some of the sets in this extension of Bell scenarios is explored in Section 5; finally, Section 6 presents some extensions of the Fine–Abramsky–Brandenburger Theorem. The paper then closes with a discussion in Section 7 and technical appendices.

## 2. Bell Nonlocality and Kochen–Specker Contextuality

### 2.1. Bell Nonlocality

A Bell scenario consists of an experimental setup where spatially separated laboratories receive physical systems from the same source and perform measurements on them. Each laboratory has a finite set of measurement settings to choose from, and it is assumed that there is no communication between different parties during the measurement processes. This procedure is iterated (in every round, each party receives a new system from the same source), and, after many runs, it is possible to estimate the probabilities for all the possible results of all possible measurements for each party. This work considers bipartite Bell scenarios, referred to as Alice and Bob, but the ideas presented here can be straightforwardly generalized to scenarios with more parties.

In the standard notion of a Bell scenario, Alice and Bob each choose only one measurement per round. Let $M_{A}$ and $M_{B}$ be the sets of possible measurements for Alice and Bob, respectively, and ${\left\{a\right\}}_{A}$ and ${\left\{b\right\}}_{B}$ be the sets of possible results given the measurements $A\in M_{A}$ and $B\in M_{B}$. Then, the Bell experiment is described using probabilities of the form:(1)p(a,b|A,B),∀a,b,A,B.
These probabilities can be organized as entries of a vector $\vec{p}\in R^{d}$, for an appropriate *d*, called the *behaviour vector*. The entries of $\vec{p}$ must all be non-negative,
(2)p(a,b|A,B)≥0,∀a,b,A,B,
and, given *A* and *B*, the associate entries must be normalized:(3)$$\sum_{a,b}p\left(a,b|A,B\right)=1,\;\forall \;A,B.$$
The first usual informational demands are the *non-signalling* conditions:(4)$$\begin{array}{cc} p(a|A) & =\sum_{b}p\left(a,b|A,B\right),\;\forall \;a,A,B,\\ p(b|B) & =\sum_{a}p\left(a,b|A,B\right),\;\forall \;b,A,B. \end{array}$$
These conditions represent, mathematically, the demand that probabilities that can be inferred by Alice (Bob) alone should not be affected by the measurement choice of Bob (Alice). The set of behaviours that satisfy non-signalling is denoted NS.

The standard notion of *locality* is that a local hidden variable (LHV) model can reproduce the correlations between the laboratories. This is expressed mathematically as
(5)$$p\left(a,b|A,B\right)=\sum_{\lambda }p\left(\lambda \right)p_{\lambda }\left(a|A\right)p_{\lambda }\left(b|B\right),\;\forall a,b,A,B,$$
where p(λ) values are the coefficients of a convex combination, while the families of probability distributions $\left\{p_{\lambda }\left(a|A\right)\right\}$ and $\left\{p_{\lambda }\left(b|B\right)\right\}$, known as *response functions*, depend only on the latent variable λ and Alice and Bob measurements, respectively. If the components of the behaviour vector $\vec{p}$ can be decomposed as Equation (Equation 5), then it is said to be *local*; otherwise, it is *nonlocal*. The set of local behaviours is denoted L. All local behaviours are also non-signalling, so L⊂NS.

The most famous and simple Bell scenario is known as the CHSH scenario [22]. In this case, the measurements of Alice and Bob are $M_{A}=\left\{A_{0},A_{1}\right\}$ and $M_{B}=\left\{B_{0},B_{1}\right\}$, respectively, and the possible results for all measurements are {−1,+1}. A comprehensive review of the field of Bell nonlocality can be found in reference [1].

### 2.2. Kochen–Specker Contextuality

A phenomenon that is related to Bell nonlocality is the so-called Kochen–Specker contextuality. Here, suppose there is a single laboratory, run by Bob, that receives physical systems and performs measurements on them. Assume, though, that in each round of the experiment, Bob is able to perform *compatible measurements* concomitantly. Each set of compatible measurements is referred to as a *context*, and the set of maximal contexts is denoted C (a context B is maximal if it is not a subset of any context). Let $B=\left(B_{1},\ldots ,B_{t}\right)\in C$ be a context of Bob and $b=\left(b_{1},\ldots ,b_{t}\right)$ a set of results, where $b_{j}\in {\left\{b\right\}}_{B_{j}}$ is a possible result of measurement $B_{j}$. The probabilities that describe the experiment are of the form
(6)p(b|B),∀b,B.
These probabilities can also be organized in a behaviour vector $\vec{p}\in R^{d}$. The non-negativity conditions,
(7)$$\begin{array}{cc} p(b|B) & \geq 0,\;\forall \;b,B, \end{array}$$
and normalization conditions,
(8)$$\begin{array}{cc} \sum_{b}p\left(b|B\right) & =1,\;\forall \;B, \end{array}$$
must be satisfied. Furthermore, it is usual to demand the *non-disturbing* condition, which is analogous to non-signalling. Given two contexts $B_{1}$ and $B_{2}$ of Bob such that their intersection $B^{\prime }=B_{1}\cap B_{2}$ is non-empty, the following must hold:(9)$$p\left(b^{\prime }|B^{\prime }\right)=\sum_{b_{1}\setminus b^{\prime }}p\left(b_{1}|B_{1}\right)=\sum_{b_{2}\setminus b^{\prime }}p\left(b_{2}|B_{2}\right),$$
where the summations are over all possible results for the measurements in the sets $B_{1}\setminus B^{\prime }$ and $B_{2}\setminus B^{\prime }$. In other words, the marginal probability distributions are well defined for all non-maximal contexts. The set of non-disturbing behaviours is denoted ND. If a behaviour does not satisfy the non-disturbing conditions, it is said to be *disturbing*.

The behaviour is said to be *non-contextual* if it can be decomposed by means of a non-contextual hidden variable (NCHV) model:(10)$$p\left(b|B\right)=\sum_{\mu }p\left(\mu \right)\prod_{j}p_{\mu }\left(b_{j}|B_{j}\right),\;\forall \;b,B,$$
where p(μ) is a probability distribution over the possible values of μ and the product is over all of *j* such that $B_{j}\in B$. The probability distributions in the set $\left\{p_{\mu }\left(b_{j}|B_{j}\right)\right\}$ are also known as response functions. If such a decomposition does not exist, then the behaviour is said to be *contextual* (it is worth mentioning that we use the term “contextual” as shorthand for “not non-contextual”). The set of non-contextual behaviours is denoted NC. Non-contextual behaviours are non-disturbing and thus, NC⊂ND.

The most famous KS scenario is the KCBS scenario [23]. For this case, the measurements of Bob are $M_{B}=\left\{B_{0},B_{1},B_{2},B_{3},B_{4}\right\}$, and the possible results for all measurements are {−1,+1}. The contexts are given by $C=\{\left(B_{0},B_{1}\right),\left(B_{1},B_{2}\right),\left(B_{2},B_{3}\right),\left(B_{3},B_{4}\right),\left(B_{4},B_{0}\right)\}$. Reference [2] provides a comprehensive review of KS contextuality.

## 3. Bell Scenarios with Compatible Measurements

In Bell scenarios, it is usually considered that each separated party performs only one measurement per round. However, a relaxation of this assumption is also possible and desirable. Ref. [18] introduces an extension of the standard Bell scenario where local compatible measurements for each party are admitted, i.e., Alice and/or Bob may measure a context. Including local compatible relations allows for the study of both Bell nonlocality and KS contextuality in a unified framework.

Let $M_{A}$ and $M_{B}$ be the sets of possible measurements for Alice and Bob, respectively, and let $C_{A}$ and $C_{B}$ be their sets of maximal contexts. Given that Alice measures a context $A=\left(A_{1},\ldots ,A_{s}\right)\in C_{A}$ and Bob measures a context $B=\left(B_{1},\ldots ,B_{t}\right)\in C_{B}$, and that $a=\left(a_{1},\ldots ,a_{s}\right)$ and $b=\left(b_{1},\ldots ,b_{t}\right)$ are sets of possible results (one for each measurement), the extended scenario is described using a set of joint conditional probabilities:(11)p(a,b|A,B),∀a,b,A,B.
As before, these probabilities can be organized as entries of a behaviour vector $\vec{p}$ and must satisfy the non-negativity and normalization conditions:(12)$$\begin{array}{cc} p(a,b|A,B) & \geq 0,\;\forall \;a,b,A,B,\\ \sum_{a,b}p\left(a,b|A,B\right) & =1,\;\forall \;A,B. \end{array}$$
These conditions describe the most general correlations in an extended Bell experiment.

In the following sections, the term *extended Bell scenarios* is used to refer to Bell experiments with compatible measurements, while the usual notion of Bell experiments is referred to as *standard Bell scenarios*. The latter is a particular case of the former, since the incompatible measurements for Alice and Bob can also be seen as maximal contexts of size one. Thus, the conditions presented in Section 2 are particular cases of what is discussed in the next sections: while standard Bell scenarios correspond to maximal contexts of size 1, KS scenarios correspond to one-party extended Bell scenarios.

### 3.1. Non-Signalling and Non-Disturbing Conditions

In many works, either the term *non-signalling* or *non-disturbing* is used to mean that the marginal probabilities for the results of one measurement are independent of the choice of other measurements that are performed concomitantly. The former is usually used for spatially separated measurements, while the latter usually refers to measurements in the same laboratory. In this work, such terms are employed with the intention of differentiating between these two situations.

The non-signalling conditions, which state that the probabilities for the results of context A of Alice do not depend on the context B chosen by Bob, and vice versa, are naturally extended:(13)$$\begin{array}{cc} p(a|A) & =\sum_{b}p\left(a,b|A,B\right),\;\forall \;a,A,B,\\ p(b|B) & =\sum_{a}p\left(a,b|A,B\right),\;\forall \;b,A,B. \end{array}$$
Behaviours that satisfy (Equation 13) belong to the non-signalling set, denoted NS.

Independently of the non-signalling conditions, the non-disturbing conditions for Bob state that for all pairs of contexts $B_{1}$ and $B_{2}$ of Bob with non-empty intersection and a subcontext $B^{\prime }$ such that $B^{\prime }\subset B_{1}$ and $B^{\prime }\subset B_{2}$, the following conditions must hold:(14)$$\begin{array}{c} p\left(b^{\prime }|A,B^{\prime }\right)=\sum_{b_{1}\setminus b^{\prime }}\sum_{a}p\left(a,b_{1}|A,B_{1}\right)=\sum_{b_{2}\setminus b^{\prime }}\sum_{a}p\left(a,b_{2}|A,B_{2}\right),\;\forall \;b^{\prime },A,B^{\prime }, \end{array}$$
where the $b_{i}$ summations of each expression are over all possible results of the measurements in $B_{1}\setminus B^{\prime }$ and $B_{2}\setminus B^{\prime }$, respectively. That is, the probabilities of the results for $B^{\prime }$ are not dependent on the context being measured. The behaviours that satisfy (Equation 14) belong to the non-disturbing set for Bob, denoted ${\mathrm{ND}}_{B}$. The analogous conditions can be valid for Alice’s measurements, and then the behaviour belongs to the non-disturbing set for Alice, denoted ${\mathrm{ND}}_{A}$. If the non-disturbing conditions are satisfied for both parties, the behaviour belongs to the Non-Disturbing set, denoted ND. Note that $\mathrm{ND}={\mathrm{ND}}_{A}\cap {\mathrm{ND}}_{B}$. Behaviours that do not satisfy the non-disturbance conditions are called *disturbing*.

As defined here, the non-signalling and non-disturbing conditions are independent. For instance, there exists the possibility of a behaviour that is non-signalling but presents disturbance, and *vice versa*. If both conditions are satisfied, the behaviour belongs to the non-signalling and non-disturbing set, denoted NS∩ND. Then, the non-disturbing conditions can be written as
(15)$$\begin{array}{cc} p(b^{\prime }|B^{\prime }) & =\sum_{b_{1}\setminus b^{\prime }}p\left(b_{1}|B_{1}\right)=\sum_{b_{2}\setminus b^{\prime }}p\left(b_{2}|B_{2}\right),\;\forall \;b^{\prime },B^{\prime }, \end{array}$$
with analogous conditions for Alice. In this case, all the marginals $p(a_{i}|A_{i})$ and $p(b_{j}|B_{j})$ are well-defined. Hence, the set NS∩ND coincides with the non-disturbing set when we look at generalized Bell scenarios as a special case of contextuality scenarios. It is also possible to combine the non-signalling conditions with non-disturbance only for Alice or Bob, resulting in the sets $\mathrm{NS}\cap {\mathrm{ND}}_{A}$ and $\mathrm{NS}\cap {\mathrm{ND}}_{B}$, respectively.

From now on, it is assumed that the behaviours considered in this article belong to the NS∩ND set.

### 3.2. Contextuality

The extension of Bell scenarios including the possibility of locally compatible measurements is suited to the study of questions related to contextuality. For this purpose, it is necessary to consider the marginal probability distributions p(a|A) and p(b|B). Written in this way, it is implicit that the non-signalling conditions (Equation 13) hold.

The behaviour $\vec{p}$ is *non-contextual* for Alice if and only if there exists a decomposition in the form:(16)$$p\left(a|A\right)=\sum_{\mu }p\left(\mu \right)\prod p_{\mu }\left(a_{i}|A_{i}\right),\;\forall \;a,A,$$
where p(μ) is a probability distribution over μ, and $\left\{p_{\mu }\left(a_{i}|A_{i}\right)\right\}$ is a set of response functions. If $\vec{p}$ does not satisfy the condition expressed in (Equation 16), then it is said to be *contextual* for Alice. The set of behaviours that are non-contextual for Alice is denoted ${\mathrm{NC}}_{A}$. In the same manner, a behaviour $\vec{p}$ is non-contextual for Bob if there is a non-contextual decomposition for Bob’s marginal behaviour p(b|B). The set of behaviours that are non-contextual for Bob is denoted ${\mathrm{NC}}_{B}$.

Finally, a behaviour $\vec{p}$ is non-contextual if and only if it is non-contextual for both Alice and Bob. Otherwise, $\vec{p}$ is contextual. The set of non-contextual behaviours is denoted NC. Note that $\mathrm{NC}={\mathrm{NC}}_{A}\cap {\mathrm{NC}}_{B}$.

The non-disturbance conditions are satisfied by every non-contextual behaviour. In extended Bell scenarios, it is necessary to specify which of the parties is non-contextual. Hence, ${\mathrm{NC}}_{A}$ is a subset of ${\mathrm{ND}}_{A}$, ${\mathrm{NC}}_{B}$ is a subset of ${\mathrm{ND}}_{B}$, and NC is a subset of ND. Moreover, since the non-signalling conditions are assumed in the definitions of all non-contextual models, the three non-contextual sets are also subsets of NS.

### 3.3. The Many Meanings of Locality in Extended Scenarios

The notion of *locality* in extended Bell scenarios is a natural extension of the standard locality condition (Equation (Equation 5)). However, it brings many new features, as is shown throughout this manuscript. A behaviour $\vec{p}$ is said to be *local* if, and only if, there exists a decomposition of the form:(17)$$p\left(a,b|A,B\right)=\sum_{\lambda }p\left(\lambda \right)p_{\lambda }\left(a|A\right)p_{\lambda }\left(b|B\right),$$
where p(λ) is a probability distribution over λ, and $\left\{p_{\lambda }\left(a|A\right)\right\}$ and $\left\{p_{\lambda }\left(b|B\right)\right\}$ are sets of response functions for Alice and Bob, respectively.

For the locality condition in (Equation 17) to be completely determined, it is necessary to specify the sets of response functions. The property of *locality* must be concerned with the possibility of a decomposition that classically correlates with the results of spatially separated laboratories; i.e., it is about finding a decomposition for the behaviour as a convex combination of probability distributions that are independent between Alice and Bob. Thus, it must be possible to have local behaviours that present contextuality in individual laboratories. To consider this situation, it is necessary to include contextual response functions in the local decomposition. Indeed, the fact that each laboratory may measure contexts opens the possibility of exploring attributes related to contextuality and the disturbance of the response functions in the local model. This brings new features to extended Bell scenarios that are not present in standard ones.

To obtain the most general definition of the local model, the only restriction that must be imposed on the sets $\left\{p_{\lambda }\left(a|A\right)\right\}$ and $\left\{p_{\lambda }\left(b|B\right)\right\}$ is that they are composed of well-defined probability distributions; in other words, they must satisfy the normalization and non-negativity conditions. This would take into account the possibility of contextual response functions. More than that, it would include both disturbing and non-disturbing response functions; i.e., they would not be constrained to satisfying the non-disturbing conditions (Equation 9). The behaviours that satisfy this locality condition, using *general response functions*, belong to the local set $L_{G}$.

As discussed above, the definition of the set $L_{G}$ includes both disturbing and non-disturbing response functions, and hence, it has both disturbing and non-disturbing behaviours. If there are non-disturbance conditions (note that the non-disturbing conditions (Equation 15) were considered instead of (Equation 14) because local behaviours satisfy the non-signalling conditions in Equation (Equation 13)), Equation (Equation 15) is imposed for both Alice and Bob on the behaviours of $L_{G}$, and the resulting set is the Local and Non-Disturbing set $L_{G}\cap \mathrm{ND}$. That is, it is ensured that, beyond having a local decomposition, the marginal behaviours of both parties are non-disturbing. It is important to emphasize that, for this case of the set $L_{G}\cap \mathrm{ND}$, disturbing response functions in the r.h.s of expression (Equation 17) are allowed, but in such a way that the probability distribution p(λ) makes the observed probabilities on the l.h.s of Equation (Equation 17) non-disturbing. It is also possible to consider non-disturbance for Alice or Bob only, resulting in the sets $L_{G}\cap {\mathrm{ND}}_{A}$ and $L_{G}\cap {\mathrm{ND}}_{B}$, respectively.

Another possibility for the definition of the locality condition (Equation 17) is to constrain the response functions $\left\{p_{\lambda }\left(a|A\right)\right\}$ and $\left\{p_{\lambda }\left(b|B\right)\right\}$ to be non-disturbing. This would define another local set, denoted by $L_{ND}$, that considers only *non-disturbing response functions*. As a convex combination of non-disturbing behaviours is also non-disturbing, it follows that the behaviours in this set automatically satisfy the non-disturbance conditions (Equation 15) for both Alice and Bob, and, hence, $L_{ND}\cap \mathrm{ND}=L_{ND}$. This was the local set considered in the original work of extended Bell scenarios [18].

Finally, the last possibility is to allow only *non-contextual response functions* in the local model. That is, for a context $A=\left(A_{1},\ldots ,A_{s}\right)$ and results $a=\left(a_{1},\ldots ,a_{s}\right)$, the response functions would be of the form
(18)$$p_{\lambda }\left(a|A\right)=\prod_{i=1}^{s}p_{\lambda }\left(a_{i}|A_{i}\right),$$
where $\left\{p_{\lambda }\left(a_{i}|A_{i}\right)\right\}$ is the response function for the measurement $A_{i}$ (in a more precise manner, the non-contextual response functions should allow for a decomposition of the form $p_{\lambda }\left(a|A\right)=\sum_{\mu }p_{\lambda }\left(\mu \right)\prod_{i=1}^{s}p_{\lambda ,\mu }\left(a_{i}|A_{i}\right)$, but this is just a convex combination of the response functions (Equation 18), which is included in the decomposition (Equation 19)). The same is valid for the response functions of Bob. The local decomposition would then be
(19)$$p\left(a,b|A,B\right)=\sum_{\lambda }p\left(\lambda \right)\left[\prod_{i=1}^{s}p_{\lambda }\left(a_{i}|A_{i}\right)\right]\left[\prod_{i=1}^{s}p_{\lambda }\left(b_{j}|B_{j}\right)\right],\;\forall \;a,b,A,B.$$

Using a marginalization process, it is possible to see that the marginal behaviours p(a|A) and p(b|B) would also have a non-contextual decomposition (see Equation (Equation 10)). Hence, with this choice of response functions, the local behaviours are also non-contextual. The local set defined using only non-contextual response functions is denoted $L_{NC}$.

Regarding the relations between these sets, it follows from the definitions that $L_{NC}$ is a subset of $L_{ND}$ and that the latter is a subset of $L_{G}$. Moreover, the two former sets are subsets of $L_{G}\cap \mathrm{ND}$, since they are also subsets of ND.

In general, the set $L_{NC}$ is a proper subset of $L_{ND}$. In scenarios where it is possible to have contextuality, there will be contextual response functions in the set of non-disturbing response functions, so the sets $L_{NC}$ and $L_{ND}$ will be different. For analogous reasons, in general, $L_{ND}$ is a proper subset of $L_{G}$, since the latter may include disturbing response functions. An interesting question is whether the sets $L_{ND}$ and $L_{G}\cap \mathrm{ND}$ are equivalent or not. In other words, is it equivalent to considering general response functions and then imposing the non-disturbance conditions or to start with non-disturbing response functions from the beginning? In Section 4.3, it is shown that, for some scenarios, these two local sets are different; i.e., $L_{ND}$ is a proper subset of $L_{G}\cap \mathrm{ND}$. Hence, these two definitions are not equivalent.

In standard Bell scenarios, where the contexts of Alice and Bob have only one measurement, these local sets coincide, i.e., $L_{NC}\equiv L_{ND}\equiv L_{G}\equiv L$.

Other local sets can be defined by choosing the different possibilities of response functions for Alice and Bob. Following the notation already introduced, let the index *NC* refer to non-contextual response functions; subscript *ND* refers to non-disturbing response functions; and *G* refers to general response functions, including both disturbing and non-disturbing ones. Then, we can also define the following sets: $L_{NC,ND}$, $L_{ND,NC}$, $L_{NC,G}$, $L_{G,NC}$, $L_{ND,G}$, and $L_{G,ND}$. In the notation $L_{I,J}$, the indices *I* and *J* refer to the allowed response functions for Alice and Bob, respectively. The local sets defined previously can also be expressed with this notation: $L_{NC}\equiv L_{NC,NC}$, $L_{ND}\equiv L_{ND,ND}$, and $L_{G}\equiv L_{G,G}$.

Note that for all defined local sets, the resultant local behaviours are non-signalling, and hence they are all subsets of NS. Also, these sets could be combined with the different conditions of non-disturbance; i.e., they can also be combined with the sets ${\mathrm{ND}}_{A}$ and ${\mathrm{ND}}_{B}$.

In this section, several local sets were defined by exploring the different features that the response functions may possess. In Section 4, the structure of these sets as convex polytopes and the minimum sufficient set of response functions for each case are discussed.

### 3.4. Locality and Non-Contextuality Together

In the previous subsections, the conditions for a behaviour to be local or non-contextual were presented separately. In this subsection, the case in which both of these properties are present is considered. For this, many combinations of the previously defined sets are possible.

First, consider the most general local set $L_{G}$ and non-contextuality at Alice’s laboratory. A behaviour $\vec{p}$ is local and non-contextual for Alice if and only if it has a local decomposition as in Expression (Equation 17), including general response functions, and a non-contextual decomposition for Alice’s marginal behaviour as in Equation (Equation 16). In other words, the behaviour must belong to the set $L_{G}\cap {\mathrm{NC}}_{A}$. Similarly, to talk about locality and non-contextuality at Bob’s laboratory, the relevant set is $L_{G}\cap {\mathrm{NC}}_{B}$. Considering non-contextuality for both Alice and Bob, a behaviour $\vec{p}$ is local and non-contextual if and only if it belongs to the set $L_{G}\cap \mathrm{NC}$. Analogous definitions are valid considering the sets $L_{ND}$ and $L_{NC}$.

From the hierarchy of the local sets discussed in the previous section, it follows that $L_{NC}\cap \mathrm{NC}$ is a subset of $L_{ND}\cap \mathrm{NC}$ and that these two are subsets of $L_{G}\cap \mathrm{NC}$. Whether these sets are equivalent is a question that has interesting consequences. To see this, note that the three sets are defined by separately imposing the local and non-contextual conditions. Nevertheless, behaviours belonging to $L_{NC}$ are automatically non-contextual, and thus, $L_{NC}\cap \mathrm{NC}=L_{NC}$. This means that for this last case, the conditions of locality and non-contextuality may be expressed by a unique hidden variable model given by Expression (Equation 19). In the case that there are behaviours belonging to $L_{ND}\cap \mathrm{NC}$ or $L_{G}\cap \mathrm{NC}$, but not to $L_{NC}$, it means that these behaviours independently have local and non-contextual decompositions; nevertheless, it is not possible to express both of them with a single model with these features. This would show that the joint consideration of these two classical concepts presents a richer structure than just having a joint global hidden variable decomposition. The question of the equivalence between these sets is answered in Section 4.3.

## 4. Geometrical Characterization of the Correlation Sets

One modern way to investigate Bell nonlocality and KS contextuality is by means of the structure of convex polytopes [1,24]. A polytope is a geometrical object that belongs to a real vector space $R^{d}$ and admits a dual characterization: it can be described either by the intersection of a finite number of half-spaces—each given by a linear inequality, related to one of its *facets*—or by the convex hull of a finite set of vectors, representing their vertices. With one of the descriptions, it is, in principle, possible to obtain the other. In practice, this problem requires an exponential amount of computational resources as the complexity of the polytope grows. In this work, the software PANDA, version 2705159a, 2016-03-27 [25], was used to find the facets or the vertices of the polytopes for the specific scenarios considered.

As previously mentioned, the probabilities that describe a scenario are organized as entries of a behaviour vector $\vec{p}\in R^{d}$. The different sets presented in Section 2 and Section 3 are defined using linear equations or linear inequalities in the entries of $\vec{p}$, or by expressing it as a convex combination of a set of response functions, which can also be organized as vectors ${\vec{p}}_{\lambda }$. Moreover, as is shown in Section 4.1, it is sufficient to consider finite sets of response functions. Hence, all the sets defined are convex polytopes.

In the following subsections, the sets of extremal vertices for each of the local sets are discussed, and the geometrical structure of convex polytopes is applied to both standard and extended Bell scenarios. By considering the facet inequalities for the polytopes of some specific extended scenarios, it was possible to prove the main results of this article.

### 4.1. Digression on Sets of Extreme Response Functions

In previous sections, the features characterizing the sets of response functions for each local set were presented. In this subsection, the minimum set of response functions that are sufficient to define these sets are discussed. These are known as *extremal response functions*, since they cannot be expressed as convex combinations of the others.

An important result is that an arbitrary probability distribution can always be written as a convex combination of deterministic probability distributions, i.e., probability distributions in which one of the results has a probability of 1. Hence, considering the local model (Equation 5) for standard Bell scenarios, the response functions $\left\{p_{\lambda }\left(a|A\right)\right\}$ and $\left\{p_{\lambda }\left(b|B\right)\right\}$ can always be expressed in terms of *deterministic response functions*. These are defined as follows: for a fixed λ, the results for the measurements $A_{i}$ and $B_{j}$ are $R_{\lambda }\left(A_{i}\right)$ and $R_{\lambda }\left(B_{j}\right)$, respectively. Hence, the extremal response functions are given by
(20)$$\begin{array}{cc} \; & p_{\lambda }\left(a_{i}|A_{i}\right)=\delta_{a_{i},R_{\lambda }\left(A_{i}\right)},\;\forall \;a_{i},A_{i},\\ \; & p_{\lambda }\left(b_{j}|B_{j}\right)=\delta_{b_{j},R_{\lambda }\left(B_{j}\right)},\;\forall \;b_{j},B_{j}. \end{array}$$

For the set $L_{NC}$, the response functions are non-contextual. Every non-contextual behaviour can be expressed as a convex combination of vertices that have deterministic results for each of the measurements. Following the notation introduced above, let $R_{\lambda }\left(A\right)=\left(R_{\lambda }\left(A_{1}\right),\ldots ,R_{\lambda }\left(A_{s}\right)\right)$ and $R_{\lambda }\left(B\right)=\left(R_{\lambda }\left(B_{1}\right),\ldots ,R_{\lambda }\left(B_{t}\right)\right)$ be tuples of determined results for the contexts $A=\left(A_{1},\ldots ,A_{s}\right)$ and $B=\left(B_{1},\ldots ,B_{t}\right)$, respectively, associated with the hidden variable λ. Then, the *deterministic and non-contextual response functions* are expressed as
(21)$$\begin{array}{cc} \; & p_{\lambda }\left(a|A\right)=\delta_{a,R_{\lambda }\left(A\right)}=\prod_{i=1}^{s}\delta_{a_{i},R_{\lambda }\left(A_{i}\right)},\;\forall \;a,A,\\ \; & p_{\lambda }\left(b|B\right)=\delta_{b,R_{\lambda }\left(B\right)}=\prod_{j=1}^{t}\delta_{b_{j},R_{\lambda }\left(B_{j}\right)},\;\forall \;b,B. \end{array}$$
These compose the set of extremal response functions for $L_{NC}$.

To find the set of extremal response functions of $L_{ND}$, it is necessary to characterize the extremal points of the non-disturbing polytopes for the marginal behaviours of Alice and Bob. As before, any non-disturbing response function can be written as a convex combination of these extremal non-disturbing points. Thus, they form the set of extremal response functions for $L_{ND}$.

Finally, for the set $L_{G}$, the only condition imposed on the response functions is that they are composed of probability distributions. As in the case of standard Bell scenarios, this means that they can be expressed in terms of deterministic vertices. However, in this case, the vertices are deterministic for each context. That is, given the context, the results of each measurement are determined, but a measurement that belongs to more than one context may have different results in each of them. This last case characterizes disturbing vertices. Note also that the set of extremal response functions of $L_{NC}$ is included in the respective set for $L_{G}$.

As can be seen in the discussion above, while in standard scenarios, the term *deterministic response function* has only one meaning, it can be ambiguous in extended scenarios: it is possible to consider deterministic and non-contextual results for each measurement or deterministic results for each context.

These sets of extremal response functions are used to find the facet inequalities for the polytopes of some specific scenarios in the next subsections.

### 4.2. Polytopes for Standard Scenarios

The characterization of standard Bell scenarios by means of convex polytopes has been known for a long time. The conditions of non-negativity, normalization, and non-signalling presented in Equations (Equation 2)– (Equation 4), respectively, define a convex polytope structure for the set of behaviours that satisfy them. This is called the *Non-signalling* polytope, NS. The local behaviours are convex combinations of a set of vectors as defined by (Equation 5). Moreover, as seen in Section 4.1, it is sufficient to consider only the deterministic response functions. This also means that local behaviours are convex combinations of a finite set of vectors, which defines another convex polytope, denominated as the *Local* polytope, L. The facet inequalities of this polytope are associated with the *Bell inequalities*. In Figure 1, there is a representation of the hierarchy of these sets for standard Bell scenarios.

For the CHSH scenario, introduced in Section 2.1, the facet inequalities of the local set L are all equivalent to the CHSH inequality (up to the relabelling of measurements and outcomes), which is, arguably, the simplest and most important Bell inequality:(22)$$\langle A_{0}B_{0}\rangle +\langle A_{0}B_{1}\rangle +\langle A_{1}B_{0}\rangle -\langle A_{1}B_{1}\rangle \leq 2,$$
with $\langle A_{i}B_{j}\rangle =p\left(a_{i}=b_{j}|A_{i},B_{j}\right)-p\left(a_{i}\neq b_{j}|A_{i},B_{j}\right)$ for i,j∈{0,1}.

The analogous reasoning is valid for the standard scenarios of contextuality. The conditions of non-negativity, normalization, and non-disturbance, given by (Equation 7)–(Equation 9), respectively, define the non-disturbing polytope, ND. The non-contextual behaviours have a non-contextual decomposition as in (Equation 10), with the set of extremal response functions being composed of the deterministic and non-contextual ones. Hence, they are convex combinations of a finite set of vectors, which defines the non-contextual polytope, NC. The facet inequalities of this polytope are associated with the Bell-like inequalities for non-contextuality.

For the KCBS scenario introduced in Section 2.2, the facet inequalities of the NC set are all equivalent (up tot the relabelling of measurements and outcomes) to the KCBS inequality:(23)$$\langle B_{0}B_{1}\rangle +\langle B_{1}B_{2}\rangle +\langle B_{2}B_{3}\rangle +\langle B_{3}B_{4}\rangle -\langle B_{4}B_{0}\rangle \leq 3,$$
with $\langle B_{j}B_{j+1\;\mathrm{mod}\;5}\rangle =p\left(b_{j}=b_{j+1\;\mathrm{mod}\;5}|B_{j},B_{j+1\;\mathrm{mod}\;5}\right)-p\left(b_{j}\neq b_{j+1\;\mathrm{mod}\;5}|B_{j},B_{j+1\;\mathrm{mod}\;5}\right)$, for j∈{0,1,2,3,4}.

### 4.3. Polytopes for Extended Scenarios

In extended Bell scenarios, the structure of convex polytopes is also present, but with a richer structure of sets. This subsection contains our main results: Theorems 1 and 2.

Due to the same reasons presented before, the non-signalling set NS and the non-disturbance sets ND, ${\mathrm{ND}}_{A}$, and ${\mathrm{ND}}_{B}$ are convex polytopes. The same holds for the combination of these conditions; i.e., NS∩ND, $\mathrm{NS}\cap {\mathrm{ND}}_{A}$, and $\mathrm{NS}\cap {\mathrm{ND}}_{B}$ are also convex polytopes. (In general, the intersection of two polytopes is also a polytope.)

One of the interests in this geometrical characterization lies in the local sets. The different sets of the extremal response functions for $L_{NC}$, $L_{ND}$, and $L_{G}$ were discussed in Section 4.1. For each of them, the local behaviours are convex combinations of a finite set of vertices, and hence they are all convex polytopes. From the sets of extremal vertices, it is possible, in principle, to find the set of facet inequalities for each of the local polytopes, which, in general, are different.

For the local and non-disturbing polytope $L_{G}\cap \mathrm{ND}$, the set of extremal vertices is not known a priori. To find them, it is first necessary to characterize the facets of $L_{G}$. Then, including the non-disturbance conditions of the set ND, the description of the $L_{G}\cap \mathrm{ND}$ set in terms of linear equations and linear inequalities is complete, and it is possible to make the reverse process and find its set of vertices. This procedure was used in the proof of Theorem 1.

As discussed in Section 3.3, the different local sets are related in the following manner:(24)$$L_{NC}\subset L_{ND}\subset L_{G}\cap \mathrm{ND}.$$
It was already stated that these sets are equivalent in standard Bell scenarios and that there are extended scenarios in which $L_{NC}\neq L_{ND}$. In the following theorem, it is shown that $L_{ND}$ is different from $L_{G}\cap \mathrm{ND}$ for some scenarios.

**Theorem** **1.**
*There are scenarios in which the set $L_{ND}$ is a proper subset of $L_{G}\cap \mathrm{ND}$.*


**Proof.** Consider a scenario where Alice has two incompatible measurements $M_{A}=\left\{A_{0},A_{1}\right\}$ and Bob has three measurements $M_{B}=\left\{B_{0},B_{1},B_{2}\right\}$ with the contexts $C_{B}=\left\{\left(B_{0},B_{1}\right),\left(B_{1},B_{2}\right)\right\}$. The possible results for all measurements are {0,1}. This is the simplest scenario where the two sets of interest may be different, since the sets $L_{ND}$ and $L_{G}$ are equal in any standard Bell scenario. In the scenario considered here, the sets $L_{NC}$ and $L_{ND}$ coincide, since all the non-disturbing response functions are also non-contextual. Following the discussion above, it is possible to find the vertices and facets of the polytopes $L_{ND}$ and $L_{G}\cap \mathrm{ND}$ [26].Table A1 presents a behaviour that is non-signalling and non-disturbing and that has a local decomposition using disturbing local boxes (see Appendix A); i.e., the behaviour belongs to $L_{G}\cap \mathrm{ND}$. However, if only non-disturbing local boxes are considered, this behaviour does not have a local decomposition, since it violates inequality (Equation 43), which is associated with a facet of $L_{ND}$. Hence, it does not belong to this set, which implies that, for this scenario, $L_{ND}\neq L_{G}\cap \mathrm{ND}$. □

The previous result implies that there are non-signalling and non-disturbing behaviours that are local if and only if disturbing local boxes are considered in the local model. This result shows that the property of non-disturbance of the boxes, which is a property related to the strategies that each individual laboratory applies locally, has implications for the global property of locality.

The behaviours that are considered in the previous theorem, which are the ones that belong to $L_{G}\cap \mathrm{ND}$ but not to $L_{ND}$, also belong to the set NS∩ND. Hence, they can be expressed as a convex combination of non-signalling and non-disturbing vertices. However, this will necessarily include nonlocal vertices. To find a convex combination of these behaviours that is also local, it is necessary to include disturbing vertices that do not belong to NS∩ND but belong to NS.

In Figure 2, a schematic representation of the hierarchy of the correlation sets for the scenario considered in the proof of Theorem 1 is presented.

The local and non-contextual sets $L_{NC},\;L_{ND}\cap \mathrm{NC},\;\mathrm{and}\;L_{G}\cap \mathrm{NC}$ are also convex polytopes. To obtain their vertices, it is first necessary to have the facet inequalities of the local polytopes and of the non-contextual polytope NC. Then, by joining the facet inequalities and the linear constraints that define each set, the vertices of the local and non-contextual polytope may be acquired.

It was stated in Section 3.4 that the behaviour sets satisfy the following relation:(25)$$L_{NC}\subset L_{ND}\cap \mathrm{NC}\subset L_{G}\cap \mathrm{NC}.\;$$
By considering the structure of convex polytopes for a specific scenario, it is possible to prove the following Theorem.

**Theorem** **2.**
*In general, the local and non-contextual sets are related in the following way:*

(26)
$$L_{NC}⊊L_{ND}\cap \mathrm{NC}⊊L_{G}\cap \mathrm{NC}.$$



**Proof.** Consider a scenario where Alice has two incompatible measurements $M_{A}=\left\{A_{0},A_{1}\right\}$ and Bob has three measurements $M_{B}=\left\{B_{0},B_{1},B_{2}\right\}$ with the contexts $C_{B}=\{\left(B_{0},B_{1}\right),$$\left(B_{1},B_{2}\right),\left(B_{2},B_{0}\right)\}$. The possible results for all measurements are {0,1}. The facet inequalities of the local sets for this scenario were obtained via facet-enumeration techniques [26]. Table A3 presents a behaviour that belongs to the set $L_{ND}\cap \mathrm{NC}$ but does not belong to $L_{NC}$, and Table A4 shows a behaviour that belongs to $L_{G}\cap \mathrm{NC}$ but does not belong to $L_{ND}\cap \mathrm{NC}$ (see Appendix B for more details). Thus, for this scenario, these sets are related as in expression (Equation 26). □

As discussed in Section 3.4, the behaviours provided in the proof of Theorem 2 have a local decomposition between the results of Alice and Bob and independent non-contextual models for the marginal behaviours of each laboratory; however, it is not possible to find a single hidden variable decomposition that encloses these two properties. A schematic representation of the relation proven in Theorem 2 is given in Figure 3.

In the next section, the set of quantum behaviours in extended Bell scenarios is defined. Its relation with the local sets is explored, which also brings some unexpected results.

## 5. Quantum Correlations

Up to this point, no physical theory has been assumed for the behaviours considered. Another important correlation set is the quantum set Q, i.e., the set of all behaviours that can be reproduced using quantum theory. In extended Bell scenarios with sets of measurements $M_{A}$ and $M_{B}$ and contexts $C_{A}$ and $C_{B}$, a behaviour $\vec{p}$ belongs to Q if there exist Hilbert spaces $H_{A}$ and $H_{B}$, a density operator ρ acting over $H_{A}⊗H_{B}$, and projective measurements $\left\{X_{a|A}\right\}$ and $\left\{Y_{b|B}\right\}$ for each $A\in M_{A}$ and $B\in M_{B}$, with the commutation relations between the quantum measurements reproducing the scenario’s contexts (that is, $\left(A_{i},A_{j}\right)\in C_{A}$ if and only if $\left[X_{a_{i}|A_{i}},X_{a_{j}|A_{j}}\right]=0$ for all results $a_{i},a_{j}$, which is also valid for Bob’s projective measurements), such that
(27)$$p\left(a,b|A,B\right)=Tr\left[\rho \left(\prod_{i=1}^{s}X_{a_{i}|A_{i}}\right)⊗\left(\prod_{j=1}^{t}Y_{b_{j}|B_{j}}\right)\right],$$
for all contexts $A=\left(A_{1},\ldots ,A_{s}\right)\in C_{A}$ and $B=\left(B_{1},\ldots ,B_{t}\right)\in C_{B}$ and results $a=\left(a_{1},\ldots ,a_{s}\right)$ and $b=\left(b_{1},\ldots ,b_{t}\right)$.

It is well known that quantum behaviours are non-signalling and non-disturbing when considering commuting projective measurements. Hence, if a behaviour $\vec{p}$ is quantum, then it also belongs to the set NS∩ND.

In standard Bell scenarios, the quantum set contains the local set. As is shown below, this is not necessarily the case for extended Bell scenarios.

In the view of generalized Bell scenarios as particular cases of contextual scenarios, the set $L_{NC}$ is equivalent to the set of non-contextual behaviours. Moreover, it is well known that all non-contextual behaviours are also quantum. Hence, it follows that $L_{NC}\subset Q$.

However, this relation does not follow for the set $L_{ND}$; i.e., considering this set, there are local behaviours that are not quantum. This is due to the possibility of post-quantum contextuality in individual laboratories.

**Theorem** **3.**
*There are scenarios in which $L_{ND}⊄Q$.*


**Proof.** Consider a scenario where Alice and Bob have measurements $M_{A}=\left\{A_{0},A_{1}\right\}$ and $M_{B}=\left\{B_{0},B_{1},B_{2},B_{3}\right\}$, respectively. The contexts are given by $C_{A}=\left\{\left(A_{0}\right),\left(A_{1}\right)\right\}$ and $C_{B}=\left\{\left(B_{0},B_{1}\right),\left(B_{1},B_{2}\right),\left(B_{2},B_{3}\right),\left(B_{3},B_{0}\right)\right\}$.On Bob’s side, the situation is analogous to a standard bipartite Bell scenario, in which each party has two dichotomic measurements. Indeed, the non-contextual inequality, in this case, is the CHSH inequality [27]. Thus, considering Bob’s non-disturbing polytope, there are contextual vertices equivalent to PR boxes, which are known to be post-quantum. Let $p_{PR}\left(b|B\right)$ be a behaviour representing one of these boxes.Consider the following joint behaviour:
(28)$$p\left(a,b|A,B\right)=p\left(a|A\right)p_{PR}\left(b|B\right),$$
where {p(a|A)} is an arbitrary behaviour for Alice. The expression (Equation 28) already gives a local decomposition with non-disturbing response functions, so this behaviour belongs to $L_{ND}$. However, due to Bob’s post-quantum marginal behaviour, it cannot be reproduced by quantum theory. Thus, $L_{ND}$ is not a subset of Q. □

The Bell inequalities of the $L_{ND}$ polytope are useful for semi-device-independent certification of entanglement, as shown in the theorem below (note that the compatibility relations necessary for the quantum contexts are not device-independent assumptions).

**Theorem** **4.**
*If a quantum state ρ is separable, then all behaviours generated from it belong to the $L_{ND}$ polytope.*


**Proof.** Since ρ is separable, there are quantum states $\left\{\rho_{A}^{\lambda }\right\}$ acting over $H_{A}$ and $\left\{\rho_{B}^{\lambda }\right\}$ acting over $H_{B}$ and a probability distribution p(λ) such that
$$\rho =\sum_{\lambda }p\left(\lambda \right)\rho_{A}^{\lambda }⊗\rho_{B}^{\lambda }.$$
Thus, a behaviour $\vec{p}$ obtained from ρ has probabilities of the form
(29)$$\begin{array}{cc} p(a,b|A,B) & =Tr\left[\rho \left(\prod_{i=1}^{s}X_{a_{i}|A_{i}}\right)⊗\left(\prod_{j=1}^{t}Y_{b_{j}|B_{j}}\right)\right]\\ \; & =\sum_{\lambda }p\left(\lambda \right)Tr\left[\left(\rho_{A}^{\lambda }⊗\rho_{B}^{\lambda }\right)\left(\prod_{i=1}^{s}X_{a_{i}|A_{i}}\right)⊗\left(\prod_{j=1}^{t}Y_{b_{j}|B_{j}}\right)\right]\\ \; & =\sum_{\lambda }p\left(\lambda \right)Tr\left[\rho_{A}^{\lambda }\prod_{i=1}^{s}X_{a_{i}|A_{i}}\right]Tr\left[\rho_{B}^{\lambda }\prod_{j=1}^{t}Y_{b_{j}|B_{j}}\right]\\ \; & =\sum_{\lambda }p\left(\lambda \right)p_{\lambda }\left(a|A\right)p_{\lambda }\left(b|B\right), \end{array}$$
where $p_{\lambda }\left(a|A\right)=Tr\left[\rho_{A}^{\lambda }\prod_{i=1}^{s}X_{a_{i}|A_{i}}\right]$ and $p_{\lambda }\left(b|B\right)=Tr\left[\rho_{B}^{\lambda }\prod_{j=1}^{t}Y_{b_{j}|B_{j}}\right]$. In this way, $\vec{p}$ is local (see Equation (Equation 17)). On the other hand, as quantum theory always generates non-disturbing behaviours, it follows that $p_{\lambda }\left(a|A\right)$ and $p_{\lambda }\left(b|B\right)$ are non-disturbing behaviours for every λ. Therefore, $\vec{p}$ belongs to $L_{ND}$. □

Since quantum behaviours in extended Bell scenarios are automatically non-disturbing, the most general local models that should be considered to test if a quantum behaviour is local or not are the ones related to the set $L_{G}\cap \mathrm{ND}$.

As expected, there are quantum behaviours that cannot be reproduced by local models, even considering general response functions. Inequality (Equation 30) is a Bell-like inequality for the $L_{G}\cap \mathrm{ND}$ polytope for the scenario of Theorem 1, which can be violated by quantum correlations [26]:(30)$$\begin{array}{cc} \; & -p\left(1,0,1|A_{0},B_{0},B_{1}\right)-p\left(1,1,0|A_{0},B_{0},B_{1}\right)-p\left(1,1,1|A_{0},B_{0},B_{1}\right)\\ \; & +p\left(1,0,1|A_{0},B_{1},B_{2}\right)+p\left(1,1,0|A_{0},B_{1},B_{2}\right)+p\left(0,0,1|A_{1},B_{0},B_{1}\right)\\ \; & +p\left(0,1,0|A_{1},B_{0},B_{1}\right)+p\left(0,1,1|A_{1},B_{0},B_{1}\right)+p\left(1,0,0|A_{1},B_{1},B_{2}\right)\\ \; & +p(1,1,1|A_{1},B_{1},B_{2})\leq 1. \end{array}$$
However, there are also behaviours in $L_{G}\cap \mathrm{ND}$ that are not quantum behaviours.

**Theorem** **5.**
*There are scenarios in which $L_{G}\cap \mathrm{ND}⊊Q$.*


**Proof.** Considering the same scenario from the proof of Theorem 1, it was verified, using the NPA hierarchy [28], that the vertex of the $L_{G}\cap \mathrm{ND}$ set presented in Table A1 is not a quantum behaviour [26]. □

This is another example of local behaviour that is not quantum. Note, however, that this post-quantum behaviour does not belong to $L_{ND}$. Moreover, for this specific scenario, neither Alice nor Bob may present contextuality (so, for this scenario, it is also valid that $L_{G}\cap \mathrm{NC}⊊Q$). Hence, the fact that quantum mechanics cannot reproduce this local behaviour is not due to post-quantum contextuality.

To conclude this section, we can state that the relation of the quantum set Q and the local set $L_{G}\cap \mathrm{ND}$ is that neither one is contained in the other. This is depicted in Figure 2, with the boundary of the quantum set represented by the dashed red line. For the specific scenario of this figure, the set $L_{ND}$ is contained inside the quantum set since it is equal to the set $L_{NC}$.

## 6. Extensions of Fine’s Theorem

Fine’s Theorem [20] is a very important result on Bell nonlocality, providing an alternative description of local behaviours. Consider a standard Bell scenario, with $M_{A}=\left\{A_{1},\ldots ,A_{n}\right\}$ and $M_{B}=\left\{B_{1},\ldots ,B_{m}\right\}$ being the sets of measurements of Alice and Bob, respectively, and ${\left\{a_{i}\right\}}_{A_{i}}$ and ${\left\{b_{j}\right\}}_{B_{j}}$ being the sets of possible results for each measurement. The scenario is described by probabilities in the form of (Equation 1). Then, a behaviour is local if and only if there exists a joint probability distribution,
(31)$$\omega \left(a_{1},\ldots ,a_{n},b_{1},\ldots ,b_{m}\right),$$
for the results of all observables involved in the scenario, such that it is possible to obtain the scenario’s probabilities from a marginalization process
(32)$$p\left(a_{i}^{\prime },b_{y}^{\prime }|A_{i},B_{j}\right)=\sum \delta_{a_{i}^{\prime },a_{i}}\delta_{b_{j}^{\prime },b_{j}}\omega \left(a_{1},\ldots ,a_{n},b_{1},\ldots ,b_{m}\right),$$
where the summation is over the possible results of all measurements. In Ref. [21], this theorem was extended to contextuality scenarios and became known as the Fine–Abramsky–Brandenburger Theorem.

Consider now an extended Bell scenario with the maximal contexts of Alice and Bob given by $C_{A}=\left\{A_{1},\ldots ,A_{p}\right\}$ and $C_{B}=\left\{B_{1},\ldots ,B_{q}\right\}$, respectively. Denote by $a_{k}$ a set of possible results for the measurements of context $A_{k}$, and by $b_{l}$ the analogous set of results for Bob. A natural question is whether Fine’s theorem generalizes to extended Bell scenarios. For the set $L_{NC}$, the theorem extends immediately.

**Theorem** **6.**
*A behaviour $\vec{p}$ belongs to $L_{NC}$ if and only if there exists a joint probability distribution for the results of all measurements, as in (Equation 31), which satisfies the condition*

(33)
$$p\left(a^{\prime },b^{\prime }|A,B\right)=\sum \delta_{a^{\prime },a}\delta_{b^{\prime },b}\omega \left(a_{1},\ldots ,a_{n},b_{1},\ldots ,b_{m}\right),\;\forall \;a^{\prime },b^{\prime },A,B$$

*where the summation is over the possible results of all measurements, and the Kronecker deltas involve only the results of the measurements in contexts A and B.*


**Proof.** Suppose that $\vec{p}$ belongs to $L_{NC}$. Then, for contexts $A=\left(A_{1},\ldots ,A_{s}\right)$ and $B=\left(B_{1},\ldots ,B_{t}\right)$, it has a local decomposition of the form
(34)$$\begin{array}{cc} p(a,b|A,B) & =\sum_{\lambda }p\left(\lambda \right)p_{\lambda }\left(a|A\right)p_{\lambda }\left(b|B\right)\\ \; & =\sum_{\lambda }p\left(\lambda \right)\left(\prod_{i=1}^{s}p_{\lambda }\left(a_{i}|A_{i}\right)\right)\left(\prod_{j=1}^{t}p_{\lambda }\left(b_{j}|B_{j}\right)\right). \end{array}$$
with the non-contextual response functions being deterministic
(35)$$\begin{array}{c} p_{\lambda }\left(a_{i}|A_{i}\right)=\delta_{R_{\lambda }\left(A_{i}\right),a_{i}},\\ p_{\lambda }\left(b_{j}|B_{j}\right)=\delta_{R_{\lambda }\left(B_{j}\right),b_{j}}, \end{array}$$
where $R_{\lambda }\left(A_{i}\right)$ is the deterministic result defined by λ for $A_{i}$, and analogously for $R_{\lambda }\left(B_{j}\right)$. Using these non-contextual response functions, it is possible to construct the following joint probability distribution, which satisfies condition (Equation 33):
(36)$$\begin{array}{c} \omega \left(a_{1},\ldots ,a_{n},b_{1},\ldots ,b_{m}\right)=\sum_{\lambda }p\left(\lambda \right)\left(\prod_{i}p_{\lambda }\left(a_{i}|A_{i}\right)\right)\left(\prod_{j}p_{\lambda }\left(b_{j}|B_{j}\right)\right), \end{array}$$
where the indices *i* and *j* in the products run over all measurements in the scenario.Suppose now that there exists a probability distribution as in (Equation 31) that satisfies Equation (Equation 33). It is possible to associate each tuple of results for all measurements with a value λ of the deterministic and non-contextual response functions:
(37)$$\left(a_{1},\ldots ,a_{n},b_{1},\ldots ,b_{m}\right)\overset{}{\to }\lambda .$$
The assignment (Equation 37) gives the deterministic result predicted by λ, i.e., $R_{\lambda }\left(A_{i}\right)=a_{i}$ and $R_{\lambda }\left(B_{j}\right)=b_{j}$. Then, using Equation (Equation 31) as a probability distribution over the values of λ and viewing the Kronecker deltas as response functions, the condition (Equation 33) can be seen as a local decomposition using only non-contextual response functions. Hence, $\vec{p}\in L_{NC}$. □

Thus, in the framework of extended Bell scenarios, the Fine–Abramsky-Branderburger Theorem characterizes the behaviours belonging to $L_{NC}$. But, as shown in Theorem 2, there are local and non-contextual behaviours that are not in this set. These behaviours demand one hidden-variable model for the results of Alice and Bob, and then, considering the marginal behaviour of each party, there are other independent hidden-variable models ensuring non-contextuality in each lab. Thus, the Fine–Abramsky–Brandenburger Theorem does not characterize all the behaviours that are both local and non-contextual.

It is also possible to extend Fine’s theorem to the set $L_{G}$. This is achieved by considering each context of the scenario as an independent measurement. In what follows, $a_{k}$ and $b_{l}$ are tuples of possible results for the contexts $A_{k}$ and $B_{l}$, respectively.

**Theorem** **7.**
*A behaviour $\vec{p}$ belongs to $L_{G}$ if and only if there exists a joint probability distribution for the results of all contexts*

(38)
$$\omega (a_{1},\ldots ,a_{p},b_{1},\ldots ,b_{q}),$$

*such that the probabilities $p(a_{k}^{\prime },b_{l}^{\prime }|A_{k},B_{l})$ of $\vec{p}$ are obtained by marginalization*

(39)
$$p\left(a_{k}^{\prime },b_{l}^{\prime }|A_{k},B_{l}\right)=\sum \delta_{a_{k}^{\prime },a_{k}}\delta_{b_{l}^{\prime },b_{l}}\omega \left(a_{1},\ldots ,a_{p},b_{1},\ldots ,b_{q}\right),$$

*where the summation is over the possible sets of results for all contexts, and the Kronecker deltas involve only the results of the contexts $A_{k}$ and $B_{l}$.*


**Proof.** Suppose that $\vec{p}$ belongs to $L_{G}$. Then, its probabilities have a local decomposition
(40)$$p\left(a_{k}^{\prime },b_{l}^{\prime }|A_{k},B_{l}\right)=\sum_{\lambda }p\left(\lambda \right)p_{\lambda }\left(a_{k}^{\prime }|A_{k}\right)p_{\lambda }\left(b_{l}^{\prime }|B_{l}\right),$$
where $\left\{p_{\lambda }\left(a_{k}^{\prime }|A_{k}\right)\right\}$ and $\left\{p_{\lambda }\left(b_{l}^{\prime }|B_{l}\right)\right\}$ are general response functions. The following probability distribution for the results of all contexts may be constructed
(41)$$\omega \left(a_{1},\ldots ,a_{p},b_{1},\ldots ,b_{q}\right)=\sum_{\lambda }p\left(\lambda \right)\left(\prod_{k=1}^{p}p_{\lambda }\left(a_{k}|A_{k}\right)\right)\left(\prod_{l=1}^{q}p_{\lambda }\left(b_{l}|B_{l}\right)\right).$$
An important remark is that if a measurement belongs to two distinct contexts, its results in each of these contexts are independent variables. With this, it is possible to verify that the property (Equation 39) is satisfied.Conversely, assume there exists a joint probability distribution (Equation 38) that satisfies the condition (Equation 39). It is possible to assign each combination of results for the contexts with a value for the summation variable:
(42)$$\left(a_{1},\ldots ,a_{p},b_{1},\ldots ,b_{q}\right)\overset{}{\to }\lambda .$$
With this assignment, the distribution (Equation 38) can be considered as a probability distribution over λ. Furthermore, the functions $\delta_{a_{k}^{\prime },a_{k}}$ and $\delta_{b_{l}^{\prime },b_{l}}$ can be viewed as deterministic response functions for a local decomposition, with the set of results determined for a specific context given by the assignment (Equation 42). Hence, condition (Equation 39) gives a local decomposition.Note that, since the results of the same measurement in different contexts should be considered independent, the response functions considered above are deterministic given the context, but it is possible to have, for the same value of λ and the same measurement, different results in different contexts; i.e., the deterministic assignment is not necessarily non-disturbing. In other words, it is necessary to consider both disturbing and non-disturbing response functions in the local model constructed in this proof. □

The final remark in the proof of Theorem 7 precludes applying the same construction to the set $L_{ND}$. It is still an open question whether there exists some version of Fine’s Theorem for this set.

## 7. Discussion

In this have manuscript, we reviewed and expanded the framework of extended Bell scenarios in which the possibility of compatible measurements for each laboratory is considered. This idea was introduced in Ref. [18], and has already led to interesting contributions to Bell nonlocality and KS contextuality [19,29].

In extended Bell scenarios, it may seem a natural idea that any behaviour that is local and locally non-disturbing can be described by a local model in which the local behaviours are non-disturbing. In this manuscript, we prove this intuitive idea wrong. Even more strikingly, we also show that local behaviours that cannot exhibit contextuality in either of their parts may also not be described by local models that use only non-contextual behaviours for each party. These results highlight the fact that even the definition of local behaviours in extended Bell scenarios is not trivial, since there is no unique straightforward extension of the one in standard Bell scenarios, and several sets of local correlations may be defined. As it happens in many fields of physics, *more is different*. In the case of generalized Bell scenarios, instead of going to more parts, we add more structure to the measurement sets and allow for more than one (compatible) measurement per part.

Another important question investigated here is the relation of the quantum set with such different local sets. It was shown that there are quantum behaviours that cannot be simulated even with the most powerful local models, as would be expected. Nevertheless, it was also shown that there are non-disturbing local behaviours, composed of disturbing local strategies, that cannot be reproduced with quantum theory using quantum models. This situation is analogous to scenarios in which it is possible to use classical communication between Alice and Bob to reproduce quantum and post-quantum behaviours [30]. In our case, the disturbance in the response functions, which allows the production of some post-quantum correlations, can be viewed as a kind of “communication” among the measurements. This may have fundamental and practical interests, e.g., in protocols that combine processing and communication, such as distributed quantum computing. Moreover, it would be interesting to explore in future works if and how sequences of incompatible quantum measurements may produce these behaviours.

Finally, different manners of defining that a behaviour is both local and non-contextual were investigated. It was shown that, in the most general way, there should be a local model regarding the locality of the measurements for the spatially separated laboratories and different independent non-contextual models for the marginal behaviours of each party. It is interesting to compare this result with the Fine–Abramsky–Brandenburger Theorem, which, for standard scenarios, characterizes the existence of a single hidden variable model for the results of all measurements, which in the framework of extended scenarios is equivalent to the set $L_{NC}$. However, as our results show (Theorem 2), there are more general ways to explain the probabilities in terms of classical models. Considering the view of Bell scenarios as a special case of contextuality scenarios, it is possible, for instance, to separate the measurements into two groups and to have a hidden-variable model that explains the behaviour as convex combinations of response functions that are independent regarding the measurements of each group. Then, it is possible to have independent hidden-variable models for the marginal behaviours of each group. This idea could be generalized to more complex groupings of the measurements, which also opens new paths to investigate in the future.

## Figures and Tables

**Figure 1 entropy-25-01276-f001:**
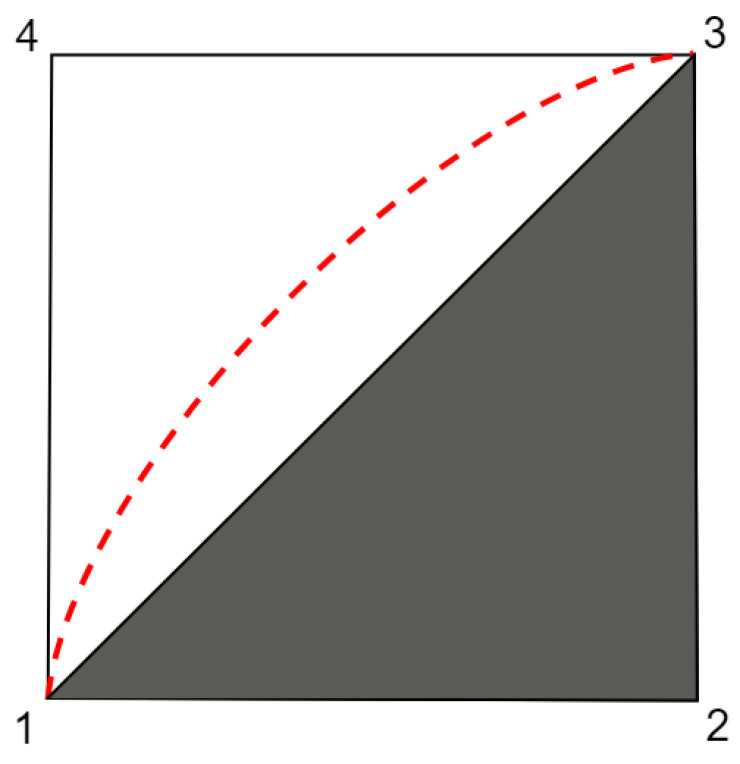
Schematic representation of the hierarchy of sets for a standard Bell scenario. The local polytope L is given by the grey polygon (1,2,3); the non-signalling polytope NS is represented by the polygon (1,2,3,4); and the quantum set is represented by the region (1,2,3) with the red dashed boundary between vertices 1 and 3.

**Figure 2 entropy-25-01276-f002:**
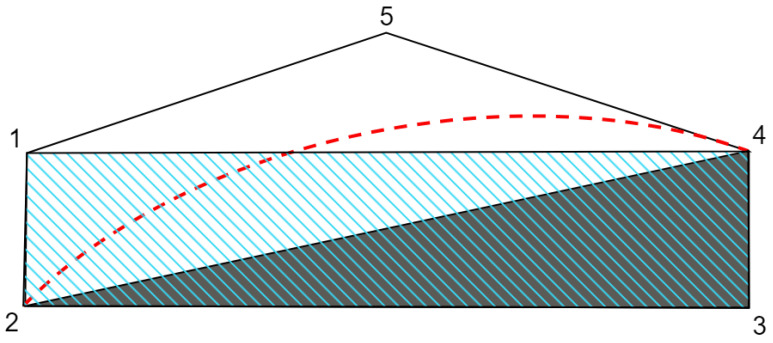
Schematic representation of the hierarchy of local sets for the scenario considered in the proof of Theorem 1. Set $L_{ND}$ is represented by the grey polygon (2,3,4); the $L_{G}\cap \mathrm{ND}$ polytope is represented by the polygon (1,2,3,4) with diagonal blue lines; the NS∩ND polytope is given by the polygon (1,2,3,4,5); the quantum set is represented by the region (2,3,4) with the red dashed boundary between vertices 2 and 4. Vertex 1 represents the behaviour in Table A1, which is a vertex of both the $L_{G}\cap \mathrm{ND}$ and NS∩ND polytopes. The common vertices of the polytopes $L_{ND}$, $L_{G}\cap \mathrm{ND}$, and NS∩ND, (2,3,4), are the deterministic and non-contextual vertices.

**Figure 3 entropy-25-01276-f003:**
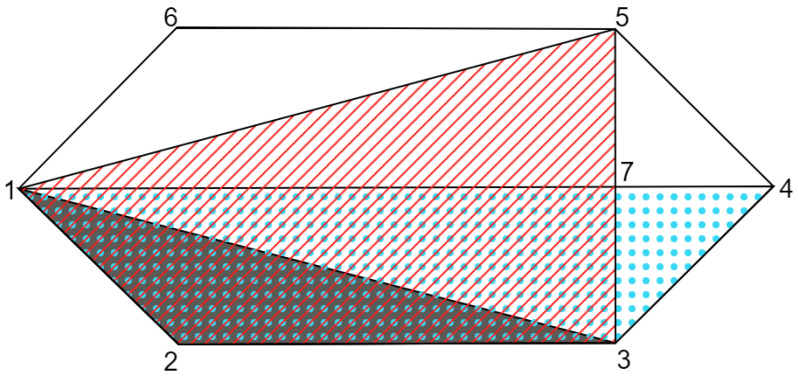
Illustration of the relation between the sets $L_{NC}$ and $L_{ND}\cap \mathrm{NC}$. The polygon with vertices (1,2,3,4,5,6) represents the NS∩ND polytope; the polygon (1,2,3,4), with blue dots, represents the set $L_{ND}$; the polygon (1,2,3,5), with diagonal red lines, represents the set NC; and the grey polygon (1,2,3) represents the $L_{NC}$. The set $L_{ND}\cap \mathrm{NC}$, represented by the polygon (1,2,3,7), with blue dots and red diagonal lines, contains the $L_{NC}$ set.

## Data Availability

Github repository. Available online: https://github.com/andremazzari/Generalized-Bell-scenarios-disturbing-consequences-on-local-hidden-variable-models, accessed on 25 August 2023.

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
