# Peer review of "Generalized Bell Scenarios: Disturbing Consequences on Local-Hidden-Variable Models"

_entropy, 2023, doi:10.3390/e25091276_

Round 1
Reviewer 1 Report
Bell nonlocality refers to the violation of Bell's inequalities, which are mathematical inequalities that impose constraints on the correlations between measurements made on entangled quantum systems. This violation implies that the behavior of these systems cannot be explained by any local hidden variable theory, where the outcomes of measurements are predetermined and independent of each other. On the other hand, Kochen-Specker contextuality refers to the property of quantum systems that their measurement outcomes depend on the context in which they are measured. This means that the outcome of a measurement cannot be determined solely by the properties of the system itself, but also depends on the other measurements being performed simultaneously. Both Bell nonlocality and Kochen-Specker contextuality challenge our classical intuitions about the nature of reality and highlight the fundamentally non-classical nature of quantum mechanics. In 3 recent works, a unified framework for these phenomena was presented. This article reviews, expands and obtains new results regarding this framework. It might be interestinf for some readers. 1. The section 2 might be devived into two subsection for different models and give some examples. 2. There are some results about the hierarchy of correlations. Can you give some examples or bell-type inequality? 3. As for quantum case, can you give some examples about quantum entanglement which can result in different correlations in your hierarchy? 4. It might be convenience for beginner with a detailed table to show their relationship.
It is good.
Author Response
Please, see the attachment.

Reviewer 2 Report
I found the paper interesting and worth publishing. It concerns a natural problem of non-locality in quantum mechanics, which in the simplest setting comprises measurements performed on two (well separated) subsystems. Here the extension is to the cases where at each laboratory several compatible measurements are done. It results of various scenarios involving contextuality and locality. Possible variants and relations among them are carefully discussed. Appealing discussions and remarks are devoted to issues of existence of hidden variable models for different scenarios.
Author Response
We thank the reviewer for the positive evaluation of our paper.